# Carbonic Anhydrase IX as a Marker of Disease Severity in Obstructive Sleep Apnea

**DOI:** 10.3390/medicina58111643

**Published:** 2022-11-14

**Authors:** Ayşegül Altıntop Geçkil, Tuğba Raika Kıran, Nurcan Kırıcı Berber, Önder Otlu, Mehmet Erdem, Erdal İn

**Affiliations:** 1Department of Chest Diseases, Malatya Turgut Özal University, Malatya 44210, Turkey; 2Department of Biochemistry, Malatya Turgut Özal University, Malatya 44210, Turkey

**Keywords:** CA-IX, intermittent hypoxia, OSA

## Abstract

*Background and Objectives:* Carbonic anhydrase (CA) enzymes are a family of metalloenzymes that contain a zinc ion in their active sites. CA enzymes have been implied in important situations such as CO_2_ transport, pH regulation, and oncogenesis. CA-IX is a transmembrane glycoprotein and stimulates the expression of hypoxia-inducible factor-1 (HIF-1) CA-IX. This study aimed to determine serum CA-IX levels in OSA patients in whom intermittent hypoxia is important and to investigate the relationship between serum CA-IX levels and disease severity. *Materials and Methods:* The study included 88 people who applied to Malatya Turgut Özal University Training and Research Hospital Sleep Disorders Center without a history of respiratory disease, malignancy, and smoking. Patients were divided into three groups: control (AHI < 5, *n* = 31), mild–moderate OSA (AHI = 5–30, *n* = 27) and severe OSA (AHI > 30, *n* = 30). The analysis of the data included in the research was carried out with the SPSS (IBM Statistics 25, NY, USA). The Shapiro–Wilk Test was used to check whether the data included in the study had a normal distribution. Comparisons were made with ANOVA in multivariate groups and the t-test in bivariate groups. ANCOVA was applied to determine the effect of the CA-IX parameter for OSA by controlling the effect of independent variables. The differentiation in CA-IX and OSA groups was analyzed regardless of BMI, age, gender, and laboratory variables. ROC analysis was applied to determine the parameter cut-off point. Sensitivity, specificity, and cut-off were calculated, and the area under the curve (AUC) value was calculated. *Results:* Serum CA-IX levels were 126.3 ± 24.5 pg/mL in the control group, 184.6 ± 59.1 pg/mL in the mild–moderate OSA group, and 332.0 ± 39.7 pg/mL in the severe OSA group. Serum CA-IX levels were found to be higher in the severe OSA group compared to the mild–moderate OSA group and control group and higher in the mild–moderate OSA group compared to the control group (*p* < 0.001, *p* < 0.001, *p* < 0.001, respectively). In addition, a negative correlation between CA-IX and minimum SaO_2_ and mean SaO_2_ (r = –0.371, *p* = 0.004; r = –0.319, *p* = 0.017, respectively). A positive correlation between CA-IX and desaturation index (CT90) was found (r = 0.369, *p* = 0.005). A positive correlation was found between CA-IX and CRP (r = 0.340, *p* = 0.010). When evaluated by ROC curve analysis, the area under the curve (AUC) value was determined as 0.940 (95% CI 0.322–0.557; *p* < 0.001). When the cut-off value for CA-IX was taken as 254.5 pg/mL, it was found to have 96.7% sensitivity and 94.8% specificity in demonstrating severe OSA. *Conclusions:* Our study found that serum CA-IX value was higher in OSA patients than in control patients, and this elevation was associated with hypoxemia and inflammation. CA-IX value can be a fast, precise, and useful biomarker to predict OSA.

## 1. Introduction

Obstructive sleep apnea (OSA) is a condition characterized by the narrowing of the upper airway during sleep [1]. In OSA, the reduction in airflow in the airway may be partial (hypopnea) or complete (apnea). As a result of all these events, hypoxemia and acidosis occur at the tissue level due to pathophysiological hypoxia–ischemia, expressed as “asphyxia”. Intermittent hypoxia is very important in the pathophysiology of OSA [2]. Transcription factors such as hypoxia-inducible factor-1 (HIF-1) are very important in OSA pathology [3,4]. It is known that inflammation occurs in the upper respiratory tract in OSA. It is known that various cytokines and acute phase proteins that show systemic inflammation, especially in advanced stages, increase in OSA, and inflammation increases as hypoxemia increases [5].

OSA affects 5–15% of the general population, and its prevalence is increasing day by day [2]. Only about 10% of individuals with OSA are diagnosed and treated. The low rate causes increased complications related to OSA [6,7]. Polysomnography is used as the gold standard for the diagnosis and classification of OSA. Today, there is a need for biomarkers to help determine OSA [8].

Carbonic anhydrase (CA) enzymes are a metalloenzyme containing a zinc ion in their active sites. It catalyzes the reactions of hydration of carbon dioxide (CO_2_) and dehydration of bicarbonate (${\mathrm{HCO}}_{3}^{-}$) alternately. CA enzyme has important physiological roles such as regulating intra and extracellular CO_2_, H^+^, and ${\mathrm{HCO}}_{3}^{-}\;$concentrations, CO_2_ transport, ion release, pH regulation, calcification, and oncogenesis [9,10]. CA has 16 isoenzymes. These isoforms may differ in catalytic activity, tissue distribution, and subcellular localization. CA has cytosolic, mitochondrial, secretory, and membrane-bound isoforms [11]. CA activity and the membrane-bound enzymes CA-IX and CA-XII increase under environmental stress conditions such as a high degree of hypoxia [12].CA-IX is a transmembrane glycoprotein and stimulates the expression of HIF-1 CA-IX [13].

Previous studies have reported that CA-IX has an important role in cancer angiogenesis, prognosis, and metastasis, and its serum levels increase in various cancer types [14]. CA activity in OSA has been investigated in treatment studies [15]. This study aimed to evaluate the relationship between CA-IX and OSA severity by examining the levels of CA-IX associated with intermittent hypoxia in patients with OSA, and to reveal the contribution of CA-IX to the pathogenesis of OSA.

## 2. Materials and Methods

### 2.1. Design of the Study and the Subjects

A total of 88 people, who applied to Malatya Turgut Özal University Training and Research Hospital Sleep Disorders Center between June and July 2022, had no respiratory disease, no history of malignancy, no smoking history, and who did not receive OSA treatment were included in the study. All patients underwent polysomnographic evaluation. Patients with OSA as a result of polysomnographic evaluation were included in the study group, and others were included in the control group. Clinical and polysomnographic measurements of the study participants were recorded. Hemoglobin, hematocrit, leukocyte count, platelet count, mean erythrocyte volume (MCV), and mean platelet volume (MPV) values of the patients’ hemogram blood parameters (SYSMEX, Automated Hematology, JAPAN) were recorded. Alanine aminotransferase (ALT), acetyl aminotransferase (AST), low-density lipoprotein (LDL), high-density lipoprotein (HDL), triglyceride, thyroid-stimulating hormone (TSH), thyrotoxin (T4), triiodothyronine (T3) and C-reactive protein (CRP) values of the patients’ biochemistry blood parameters (ARCHITECT, Toshiba, Abbott Park, USA) were recorded. Blood samples were taken from patients with OSA after completing the anamnesis and physical examination.

### 2.2. Polysomnographic Evaluation

Patients were evaluated using a 55-channel polysomnography (Alice 6^®^ Sleepware, Philips Respironics, PA, USA) system. Polysomnography recordings were analyzed according to the American Academy of Sleep Medicine (AASM) criteria version 2.6 [6]. Apnea was defined as a drop in peak thermal sensor excursion by ≥90% of baseline for ≥10 s. Hypopnea was defined as a drop in the nasal pressure signal by ≥30% of baseline for ≥10 s, causing a ≥3% decrease in the oxygen saturation (SaO_2_) pre-event baseline or arousal. The desaturation index (CT90) was defined as the number of oxygen desaturations per hour during sleep. The minimum SaO_2_ was defined as the lowest SaO_2_ value recorded during the night. Mean SaO_2_ was defined as the average SaO_2_ value recorded during the night. The apnea–hypopnea index (AHI) was defined as the mean number of apneic and hypopneic episodes per hour of sleep. An obstructive AHI > 5/h was considered diagnostic of OSA. The patients were categorized into mild (AHI, 5–15), moderate (AHI, 15–30), and severe OSA (AHI, >30) groups. Patients with AHI < 5 were considered as the control group.

### 2.3. Determination of CA-IX Levels in Serum Samples

Venous blood samples taken from the cubital vein for the study were centrifuged at 2000 RPM and stored at −80 °C. The CA-IX value was analyzed according to kit procedures using the commercial enzyme-linked immune assay [Enzyme-Linked Immuno Sorbent Assay (ELISA)] kit (Cloud-Clone Corp., Cat. No: SED076Hu, Wuhan, China). The absorbance of the samples was determined with a microplate reader adjusted to 450 nm wavelength. The measurement range was 7.81–500 pg/mL, while the minimum measurable level was <2.9 pg/mL.

### 2.4. Statistical Analysis

The analysis of the data included in the research was carried out with the SPSS (IBM Statistics 25, New York, NY, USA). The Shapiro–Wilk Test was used to check whether the data included in the study had a normal distribution. Comparisons were made with ANOVA in multivariate groups and the t-test in bivariate groups. ANCOVA was applied to determine the effect of the CA-IX parameter for OSA by controlling the effect of independent variables. The differentiation in CA-IX and OSA groups was analyzed regardless of BMI, age, gender, and laboratory variables. ROC analysis was applied to determine the parameter cut-off point. Sensitivity, specificity, and cut-off were calculated, and the area under the curve (AUC) value was calculated.

## 3. Results

### 3.1. Basic Demographic and Polysomnographic Data

Patients were divided into three groups: control (AHI < 5, *n =* 31), mild–moderate OSA (AHI = 5–30, *n =* 27), and severe OSA (AHI > 30, *n =* 30). Thirty-five (39.8%) of the patients were female, and 53 (60.2%) were male, and there was a significant difference between the groups in terms of gender (*p* < 0.001). The mean age of the patients was 46.38 ± 11.79 years, and there was a significant difference between the groups in terms of age (*p* = 0.03). The mean BMI of the patients was 28.75 ± 5.79, and there was a significant difference between the groups in terms of BMI (*p* < 0.001). While there was a history of hypertension in 7 people in the mild–moderate OSA group and 17 people in the severe OSA group, there was no hypertension patient in the control group. The patients did not have a regular history of antihypertensive drug use.

The mean AHI value in the polysomnography evaluation was 41.03 ± 24.91. The mean SaO_2_ value of the patients was 90.58 ± 3.16, and there was a significant difference between the groups (*p* = 0.002). The mean minimum SaO_2_ was 77.14 ± 11.67, and there was a significant difference between the groups (*p* < 0.001). The mean CT90 was 25.15 ± 29.80, and there was a significant difference between the groups (*p* < 0.001) (Table 1).

### 3.2. Evaluation of Laboratory Levels and CA-IX Levels

Serum CA-IX levels were 126.3 ± 24.5 pg/mL in the control group, 184.6 ± 59.1 pg/mL in the mild–moderate OSA group, and 332.0 ± 39.7 pg/mL in the severe OSA group. Serum CA-IX levels were significantly higher in the severe OSA group compared to the mild–moderate OSA group and the control group (*p* < 0.001, *p* < 0.001, respectively). Serum CA-IX levels were also found to be statistically significantly higher in the mild–moderate OSA group compared to the control group (*p* < 0.001) (Table 1, Figure 1). A significant difference was found between the control group and severe OSA regarding hemoglobin and hematocrit values, which are hemogram parameters (*p* = 0.05, *p* = 0.02). There was no difference between the groups regarding leukocyte count, platelet count, MPV, and MCV values. There was no significant difference between the control group and severe OSA in terms of LDL and triglyceride values, which are biochemistry parameters (*p* = 0.023, *p* = 0.026). A significant difference was found between the control group and severe OSA regarding CRP value (*p* = 0.022). There was no difference between the groups regarding AST, ALT, HDL, TSH, T3, and T4 values (Table 2). CA-IX values did not differ in OSA groups according to the presence of hypertension (*p* > 0.05).

When the effects of BMI, age, gender, hemoglobin, hematocrit, LDL, triglyceride, and CRP variables were fixed using ANCOVA, the CA-IX variable did not differ in the OSA groups. It was observed that 63.1% (η^2^ = 0.631, *p* = 0.001) of the change in OSA groups was caused by CA-IX. When the intragroup CA-IX differences were examined, a statistically significant difference was found between the control group and severe OSA and between the mild–moderate OSA and the severe OSA group (*p* < 0.05). There was no significant difference between the control group and the mild–moderate OSA group (*p* > 0.05) (Table 3).

### 3.3. Correlation Analysis

There was a negative correlation between serum CA-IX levels and minimum SaO_2_ and mean SaO_2_ (r = −0.371, *p* = 0.004; r = −0.319, *p* = 0.017, respectively). There was a positive correlation (r = 0.369, *p* = 0.005) between serum CA-IX levels and CT90. There was a positive correlation between CRP and CA-IX values (r = 0.340, *p* = 0.01).

### 3.4. ROC Curve Analysis

When evaluated by ROC curve analysis, the area under the curve (AUC) value was determined as 0.940 (95% CI 0.322–0.557; *p* < 0.001). When the cut-off value for CA-IX was taken as 254.5 pg/mL, it was found to have 96.7% sensitivity and 94.8% specificity in demonstrating severe OSA (Figure 2).

## 4. Discussion

The results of our study conducted to determine serum CA-IX levels in OSA patients show that serum CA-IX levels are significantly higher in both mild–moderate and severe OSA patients compared to control patients. Serum CA-IX levels were negatively correlated with minimum SaO_2_ and mean SaO_2_ values and positively correlated with CT90. Serum CA-IX levels showed a positive correlation with the CRP value.

CA-IX, a membrane-associated CA strongly induced by hypoxia, is particularly overexpressed in some types of cancer and has been associated with metastasis and poor prognosis [16]. Tumor-associated CA-IX has two main forms. One is the form expressed in gastric mucosa and various types of cancer, and the other is CA-IX, which is released from transmembrane and intracellular arrays by cleavage with proteolytic enzymes [17]. Studies have shown that CA-IX contributes to the acidification of hypoxic tumors by reducing the extracellular pH (<7.28) [18]. Low pH has been associated with extracellular matrix degradation, tumor transformation, and tumor cell migration [19]. In renal cell carcinoma, unlike other tumors, CA-IX overexpression has been associated with a good prognosis and a good response to immunotherapy [20]. Inhibition of CA-IX release in these patients has been shown to reduce tumor proliferation and tumor resistance to conventional anti-cancer therapies [20]. It has also been found that acetazolamide, a CA inhibitor, prevents tumor invasion [16].

CA enzymes are abundant in the human body and have also been associated with cardiovascular diseases [21]. The enzymatic function of CA is essential for maintaining acid–base balance and, therefore, controlling respiration. High CA activity in OSA has been clinically investigated by therapy with drugs with CA-inhibiting properties [15]. Recent studies have shown that pharmacological blockade of the CA enzyme by acetazolamide improves both blood pressure and sleep apnea [22]. It has also been reported that using zonisamide, a CA inhibitor, reduces blood pressure in OSA patients [23]. Acetazolamid has also been used to prevent hypertension in altitude sickness [24].

When the studies on OSA in the literature were examined, Wang et al. found that patients with OSA and concomitant hypertension had higher blood CA activity than OSA patients without hypertension. It was observed that OSA and blood pressure were controlled, and CA activity decreased by inhibition of CA with acetazolamide [25]. In another study, long-term positive airway pressure (PAP) treatment was found to improve OSA and blood pressure, but it was observed that CA activity did not change [12]. In a prospective study conducted on patients with OSA and hypertension, CPAP treatment was given to one group, and acetazolamide treatment was given to the other group in addition to CPAP treatment, and the results were compared after two weeks. At the end of the study, it was found that there was a significant improvement in AHI, CT90, mean SaO_2_, and minimum SaO_2_ in patients who were given additional acetazolamide [23]. The improvement in hypoxia-related parameters can be explained by the reduction of loop gain in OSA by acetazolamide and the decrease in ${\mathrm{HCO}}_{3}^{-}\;$ levels following acetazolamide treatment [26].

In our study, in which we investigated CA-IX levels, a parameter associated with tissue hypoxemia in OSA patients, we found a correlation between OSA severity and CA-IX. When the cut-off value for CA-IX was taken as 254.5 pg/mL, we found that it had 96.7% sensitivity and 94.8% specificity in demonstrating severe OSA. It showed that these results were independent of hypertension and that hypertension did not affect CA-IX levels among the OSA groups. When the effects of age, gender, BMI, hemogram, and biochemistry parameters were fixed in covariance analysis, we found that the CA-IX value differed between OSA grades. We concluded that this difference between the groups was associated with hypoxemia and inflammation. The negative correlation between CA-IX levels and minimum SaO_2_ and mean SaO_2_ values and the positive correlation with CT90 also show the relationship with hypoxia. The positive correlation between CA-IX and CRP shows the relationship with inflammation; as hypoxemia increases in OSA, inflammation also increases.

The limitation of our study was the relatively small number of OSA and control patients and the lack of CA-IX evaluation of OSA patients after PAP treatment. CA-IX may be a promising parameter in diagnosing and treating OSA and other respiratory tract diseases in which hypoxemia is at the forefront. More extensive studies are needed to determine the importance of CA-IX and to support our findings.

## 5. Conclusions

In conclusion, this study showed that serum CA-IX levels are increased in severe OSA patients, which is associated with hypoxemia and inflammation. CA-IX can be a fast, precise, and useful biomarker to predict patients at high risk of OSA.

## Figures and Tables

**Figure 1 medicina-58-01643-f001:**
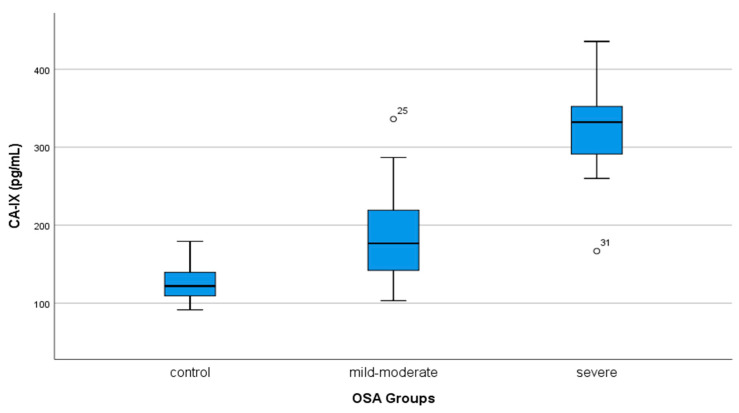
The comparison of serum CA IX levels in OSA groups.

**Figure 2 medicina-58-01643-f002:**
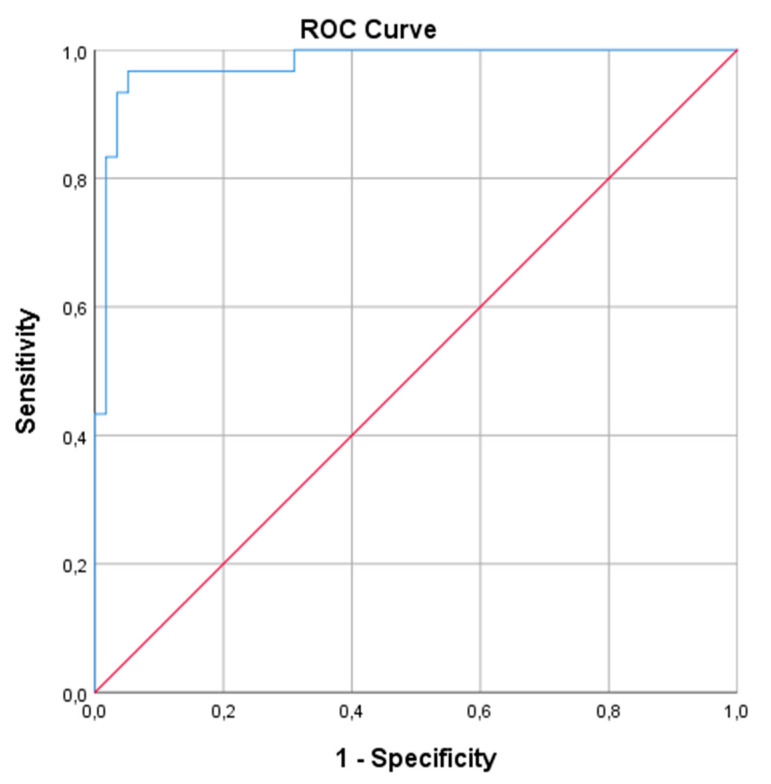
ROC curve analysis of the utility of CA IX to severe OSA. The red line is the reference and the blue line is the ROC curve.

**Table 1 medicina-58-01643-t001:** Evaluation of basic demographic and polysomnographic data.

Variable	OSA Group	Mean	Std. Deviation	F	*p*	Difference
Age (year)	Control ^1^	41.06	9.40	8.577	<0.001	1 and 3
	Mild–moderate ^2^	47.78	11.62			
	Severe ^3^	53.57	13.98			
BMI (kg/m^2)^	Control ^1^	24.63	2.33	20.634	<0.001	1–2, 1–3, 2–3
	Mild–moderate ^2^	29.41	5.51			
	Severe ^3^	32.53	5.92			
AHI (/h)	Control ^1^	3.52	3.24	58.576	<0.001	1–2, 1–3, 2–3
	Mild–moderate ^2^	19.71	7.22			
	Severe ^3^	60.70	18.52			
Minimum SPO_2_ (%)	Control ^1^	92.00	5.65	10.118	<0.001	1–2, 1–3, 2–3
	Mild–moderate ^2^	83.33	3.88			
	Severe ^3^	71.20	13.57			
Mean SPO_2_ (%)	Control ^1^	93.00	6.52	7.222	0.002	1–2, 1–3, 2–3
	Mild–moderate ^2^	92.00	1.41			
	Severe ^3^	89.14	3.75			
CT90 (/h)	Control ^1^	0.30	0.24	10.189	<0.001	1–2, 1–3, 2–3
	Mild–moderate ^2^	9.81	15.60			
	Severe ^3^	40.30	32.60			

ANOVA Test value, *p* < 0.05; There is a difference between groups, ^1^ AHI < 5; ^2^ AHI, 5–30; ^3^AHI > 30.

**Table 2 medicina-58-01643-t002:** Evaluation of laboratory levels and CA-IX levels.

Variable	OSA Group	Mean	Std. Deviation	F	*p*	Difference
CA-IX (pg/mL)	Control ^1^	126.58	24.93	149.271	0.000 *	1–2, 1–3, 2–3
	Mild–moderate ^2^	184.60	59.07			
	Severe ^3^	324.43	48.69			
HDL (mg/dL)	Control ^1^	44.54	7.43	0.178	0.838	none
	Mild–moderate ^2^	43.85	9.75			
	Severe ^3^	45.79	12.98			
LDL (mg/dL)	Control ^1^	89.31	24.40	4.052	0.023 *	1 and 3
	Mild–moderate ^2^	97.35	31.67			
	Severe ^3^	119.29	34.53			
TG (mg/dL)	Control ^1^	128.10	41.77	3.940	0.026 *	1 and 3
	Mild–moderate ^2^	178.00	87.24			
	Severe ^3^	232.08	128.95			
AST (U/L)	Control ^1^	14.00	0.5	1.030	0.365	none
	Mild–moderate ^2^	19.48	5.05			
	Severe ^3^	18.95	5.46			
ALT (U/L)	Control ^1^	10.50	2.12	1.314	0.278	none
	Mild–moderate ^2^	22.92	7.57			
	Severe ^3^	25.29	16.61			
T3 (pg/mL)	Control ^1^	3.19	0.72	0.414	0.664	none
	Mild–moderate ^2^	3.41	0.38			
	Severe ^3^	3.31	0.47			
T4 (ng/mL)	Control ^1^	1.05	0.02	1.013	0.371	none
	Mild–moderate ^2^	1.24	0.17			
	Severe ^3^	1.23	0.18			
TSH (mU/L)	Control ^1^	2.30	0.84	0.757	0.475	none
	Mild–moderate ^2^	1.70	0.53			
	Severe ^3^	2.03	1.35			
Hemoglobin (g/dL)	Control ^1^	13.78	2.02	5.617	0.005 *	1 and 3
	Mild–moderate ^2^	14.87	1.64			
	Severe ^3^	15.33	1.49			
Hematocrit (%)	Control ^1^	40.93	5.48	6.796	0.002 *	1 and 3
	Mild–moderate ^2^	44.07	4.51			
	Severe ^3^	45.81	4.64			
Leukocyte 10^3^/L	Control ^1^	7.50	1.69	1.559	0.217	none
	Mild–moderate ^2^	7.33	1.58			
	Severe ^3^	8.15	1.85			
Thrombocyte 10^3^/L	Control ^1^	289.30	63.11	2.107	0.129	none
	Mild–moderate ^2^	251.73	67.10			
	Severe ^3^	280.45	73.44			
MCV (fL)	Control ^1^	85.18	4.81	0.022	0.979	none
	Mild–moderate ^2^	85.20	4.90			
	Severe ^3^	85.45	5.39			
MPV (fL)	Control ^1^	10.28	0.81	0.043	0.958	none
	Mild–moderate ^2^	10.21	0.86			
	Severe ^3^	10.25	0.94			
CRP (mg/dL)	Control ^1^	0.21	0.18	4.100	0.022 *	1 and 3
	Mild–moderate ^2^	0.36	0.37			
	Severe ^3^	0.71	0.68			

ANOVA Test value; * *p* < 0.05; There is a difference between groups, ^1^ AHI < 5, ^2^ AHI, 5–30, ^3^ AHI > 30.

**Table 3 medicina-58-01643-t003:** Effect of CA-IX on OSA groups when the effect of variables is standardized.

Variable	F	*p*	η^2^
BMI	0.407	0.528	0.013
Age	0.033	0.857	0.001
Gender	0.464	0.501	0.015
LDL	0.377	0.544	0.012
TG	0.021	0.885	0.001
Hemoglobin	0.095	0.760	0.003
Hematocrit	0.001	0.977	0.001
CRP	0.033	0.857	0.001
OSA Group	25.732	0.001 *	0.632
CA IX	Control ^1^	Mild–moderate ^2^	Severe ^3^
Mean ± ss	124.99 ± 22.46	172.28 ± 14.84	327.81 ± 16.77
Difference	1–3, 2–3		

CA-IX, dependent variable; ss, standard deviation; η^2^, partial Eta squared; F, ANCOVA test value; * *p* < 0.05; There is a difference between groups, ^1^ AHI < 5; ^2^ AHI, 5–30; ^3^ AHI > 30.

## Data Availability

The datasets are not publicly available but are available from the corresponding author upon reasonable request.

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
