# Peer review of "Carbonic Anhydrase IX as a Marker of Disease Severity in Obstructive Sleep Apnea"

_medicina, 2022, doi:10.3390/medicina58111643_

Round 1

Reviewer 1 Report

Dear Authors,

I have some minor comments regarding your manuscript. I recommend modifying them before further reviewing. I hope my comments can help improve the quality of this study.

Best regards,

===========

Comment 1: Line 67-"The relationship between OSA and CA-IX has not been studied before in the literature." is not correct. At least, I find this article is related to your manuscript (Evidence of Placental Hypoxia in Maternal Sleep Disordered Breathing. Pediatr Dev Pathol. 2015 Sep-Oct;18(5):380-6.). Please modify this sentence.

Comment 2: Line 96- "An AHI> 5/h was considered diagnostic of OSA." should be modified to "An obstructive AHI > 5/h was considered diagnostic of OSA." Exactly, the corresponding categorization of two OSA groups may need to be changed. Or the topic might be changed to "sleep apnea" from "obstructive sleep apnea"

Comment 3: Line 180- "When evaluated by ROC curve analysis, the area under the curve (AUC) value was determined as 0.440 (95% CI 0.322-0.557; p<0.001)." Why is AUC not greater than 0.500? According to the ROC curve, I suppose the AUC will be 0.940.

Comment 4: Line 119: How to determine the sample size is efficient? As the study limitations are acknowledged, the results are still preliminary and lack external validation. Please turn down the tone of the conclusions because the present study needs further validation.

Author Response

Comment 1: Line 67-"The relationship between OSA and CA-IX has not been studied before in the literature." is not correct. At least, I find this article is related to your manuscript (Evidence of Placental Hypoxia in Maternal Sleep Disordered Breathing. Pediatr Dev Pathol. 2015 Sep-Oct;18(5):380-6.). Please modify this sentence.

Line 67: This sentence was removed from the study.

Comment 2: Line 96- "An AHI> 5/h was considered diagnostic of OSA." should be modified to "An obstructive AHI > 5/h was considered diagnostic of OSA." Exactly, the corresponding categorization of two OSA groups may need to be changed. Or the topic might be changed to "sleep apnea" from "obstructive sleep apnea"

Line 96: Change has been made.

Comment 3: Line 180- "When evaluated by ROC curve analysis, the area under the curve (AUC) value was determined as 0.440 (95% CI 0.322-0.557; p<0.001)." Why is AUC not greater than 0.500? According to the ROC curve, I suppose the AUC will be 0.940.

Line 180: There is a typo. The AUC value was written as 0.940.

Comment 4: Line 119: How to determine the sample size is efficient? As the study limitations are acknowledged, the results are still preliminary and lack external validation. Please turn down the tone of the conclusions because the present study needs further validation.

 Sample size calculated by Power G analysis. The power of the study was calculated as 95.4%

Reviewer 2 Report

This study by Geckil and coworkers deals with the serum levels of the carbonic anhydrase (CA) CA-IX concentration in 88 patients recruited to a hospital training and research center. Patients were subdivided into those without obstructive sleep apnea (OSA, AHI<5), those with Mild OSA (AHI 5-30) and patients with severe OSA (AHI>30). CA-IX increased in a curvilinear fashion across CA-IX classes. Several metrics of hypoxia were linearly associated with CA-IX. As expected, age, BMI, AHI, LDL, TG, Hb, Hct and CRP increased across AHI severity class. A CA-IX cut of approximately 250 pg/ml produced a 96.7% sensitivity and 94.8% specificity to predict severe OSA and was proposed as a useful biomarker in OSA.

Overall, this is an interesting study on a relevant and current topic. The findings have implications for identification of biomarkers and therapies in OSA. However, there are both major and minor issues that need to addressed in the current version.

Major comments

There is a need for a better description of the patient cohort in this study. The current classification of a control group is unclear. Were subjects in the lowest AHI class recruited into the study due to symptoms of OSA or previous treatment experience in OSA? How were subjects in this study selected?

Why 88 patients? Relevant power computation?

Was this this a retrospective analysis cohort or did you formulate a hypothesis prior to study start?

Was there an application filed for an Ethics Review board assessment?

Was the study registered in ClinicalTrials.gov, EudraCT or similar?

Minor comments

Line 13. Change to: . ..“have been implied in”…

Line 21 Move detailed statistics section to the method section

Line 67: The statement that CA-IX has not been studied before is formally correct, but it should be recognized that the principle of CA activity in OSA is described in refs 12 and 24 which should be cited here. In fact, the concept of high CA activity in OSA has been clinically explored by therapy with drugs with CA-inhibitory properties such as described in references 21, 22 and 26. Yet an important reference on this topic which appears relevant and should be added is: Am J Respir Crit Care Med 2022 Jun 15;205(12):1461-1469. doi: 10.1164/rccm.202109-2043OC. Finally, references 12 and 25 are double citations.

Line 77: Was the central laboratory certified?

Line 92 Better clarification of “desaturation index” needed. Current description lacks time domain. Pls look into the literature to get the correct description.

Line 133 Pls check mean values for AHI events in mild/moderate group. Typo? If AHI should be <5 how could the control value be 26.6? Only one decimal in table.

Line 149. This is interesting difference with respect to other studies. Treated hypertension? Use of diuretics?

Line 152: Figure nice. Add figure text to describe what we see. Give concentration of CA-IX on y-axis (pg/ml)

Line 153: Table 2 and Table 3 are OK. However, Table 3 is not clear with respect to details of method used. Drop “there is difference between groups” in table text

Line 170 (and on): Figures 2, 3, 4 and 5 may all be omitted according to this reviewer. Also, the population is evidently skewed. Single extreme values govern the correlation analysis. Findings or correlations (if used) may be described in running text. In Figure 2, how can you assess a minimum SpO2 of 35-45. Oximeters are rarely reliable in this range.

Line 208 Discussion reads well but pls include AJRCCM reference cited above

Author Response

Major comments

There is a need for a better description of the patient cohort in this study. The current classification of a control group is unclear. Were subjects in the lowest AHI class recruited into the study due to symptoms of OSA or previous treatment experience in OSA? How were subjects in this study selected?

Materials and methods:  A total of 88 people, who applied to Malatya Turgut Özal University Training and Research Hospital Sleep Disorders Center between June and July 2022, had no respiratory disease, no history of malignancy, no smoking history, and who did not receive OSA treatment were included in the study.

The sentence ‘All patients underwent polysomnographic evaluation. Patients with OSA as a result of polysomnographic evaluation were included in the study group, and others were included in the control group’’ added to ‘’materials and  methods’’ section .

The sentence ''Patients with AHI<5 were considered as the control group'' added to the end of the "Polysomnographic Evaluation".   

Why 88 patients? Relevant power computation?

Sample size calculated by Power G analysis. The power of the study was calculated as 95.4%

Was this this a retrospective analysis cohort or did you formulate a hypothesis prior to study start?

This study is a hypothesis study, not a retrospective analysis.

Was there an application filed for an Ethics Review board assessment?

Approval was obtained from Malatya Turgut Özal University Clinical Research Ethics Committee (issue:2022/25).

Was the study registered in ClinicalTrials.gov, EudraCT or similar?

No

Minor comments

Line 13. Change to: . “have been implied in”.

Change has been made.

Line 21 Move detailed statistics section to the method section

Suggested additions has been made.

Line 67: The statement that CA-IX has not been studied before is formally correct, but it should be recognized that the principle of CA activity in OSA is described in refs 12 and 24 which should be cited here. In fact, the concept of high CA activity in OSA has been clinically explored by therapy with drugs with CA-inhibitory properties such as described in references 21, 22 and 26. Yet an important reference on this topic which appears relevant and should be added is: Am J Respir Crit Care Med 2022 Jun 15;205(12):1461-1469. doi: 10.1164/rccm.202109-2043OC. Finally, references 12 and 25 are double citations.

CA activity in OSA has been investigated in treatment studies ‘’ added to INTRODUCTION section

Reference number 25 has been changed as #12.

Line 77: Was the central laboratory certified?

Yes.   Sleep Disorders Center is accredited.  For hemogram blood parameters (SYSMEX, Automated Hematology, JAPAN), for biochemistry blood parameters (ARCHITECT, Toshiba, Abbott Park, USA) added.

Line 92 Better clarification of “desaturation index” needed. Current description lacks time domain. Pls look into the literature to get the correct description.

Desaturation index (CT90) was defined as the number of oxygen desaturations per hour during sleep.

Line 133 Pls check mean values for AHI events in mild/moderate group. Typo? If AHI should be <5 how could the control value be 26.6? Only one decimal in table.

There is a typo. The AHI value was written as 3.52.

Line 149. This is interesting difference with respect to other studies. Treated hypertension? Use of diuretics?

The patients did not have a regular history of antihypertensive and diuretic drugs use.

Line 152: Figure nice. Add figure text to describe what we see. Give concentration of CA-IX on y-axis (pg/ml)

Recommended changes has been made.

Line 153: Table 2 and Table 3 are OK. However, Table 3 is not clear with respect to details of method used. Drop “there is difference between groups” in table text

‘’ANOVA Test value, *p<0.05; There is difference between groups, 1 AHI<5, 2 AHI, 5–30, 3AHI>30.’’ added.

Line 170 (and on): Figures 2, 3, 4 and 5 may all be omitted according to this reviewer. Also, the population is evidently skewed. Single extreme values govern the correlation analysis. Findings or correlations (if used) may be described in running text. In Figure 2, how can you assess a minimum SpO2 of 35-45. Oximeters are rarely reliable in this range.

Figures 2, 3, 4, 5 were omitted from the study because findings or correlations were described in the text flow.

Line 208 Discussion reads well but pls include AJRCCM reference cited above

‘’High CA activity in OSA has been clinically investigated by therapy with drugs with CA-inhibiting properties (Am J Respir Crit Care Med 2022 Jun 15;205(12):1461-1469)’’    added to DISCUSSION section